# INSTANCE-AWARE GRAPH PROMPT LEARNING

## ABSTRACT

Graph neural networks stand as the predominant technique for graph representation learning owing to their strong expressive power, yet the performance highly depends on the availability of high-quality labels in an end-to-end manner. Thus the pretraining and fine-tuning paradigm has been proposed to mitigate the label cost issue. Subsequently, the gap between the pretext tasks and downstream tasks has spurred the development of graph prompt learning which inserts a set of graph prompts into the original graph data with minimal parameters while preserving competitive performance. However, the current exploratory works are still limited since they all concentrate on learning fixed task-specific prompts which may not generalize well across the diverse instances that the task comprises. To tackle this challenge, we introduce *Instance-Aware Graph Prompt Learning* (IA-GPL) in this paper, aiming to generate distinct prompts tailored to different input instances. The process involves generating intermediate prompts for each instance using a lightweight architecture, quantizing these prompts through trainable codebook vectors, and employing the exponential moving average technique to ensure stable training. Extensive experiments conducted on multiple datasets and settings showcase the superior performance of IA-GPL compared to state-of-the-art baselines.

## 1 INTRODUCTION

Graphs function as pervasive data structures employed across various real-world applications, including but not limited to social networks Guo & Wang (2020); Liu et al. (2021c), molecular structures Mercado et al. (2021); Guo et al. (2021), and knowledge graphs Liu et al. (2021a); Ye et al. (2022), due to their efficacy in modeling intricate relationships. With the rise of deep learning, Graph Neural Networks (GNNs) have emerged as a formidable technique for analyzing graph data.

Nevertheless, GNNs trained end-to-end exhibit a strong dependency on large-scale high-quality labeled data for supervision, which can be challenging or costly to obtain in real-world scenarios. To overcome this challenge, researchers have explored self-supervised or pre-trained GNNs Zhu et al. (2021); Jin et al. (2020); Xia et al. (2022); You et al. (2020a) inspired by the advancements in vision Fan et al. (2021) and language Bao et al. (2021) domains. The pre-training methodologies using readily accessible label-free graphs aim to capture intrinsic graph properties (e.g., node features, node connectivity, or sub-graph pattern) that exhibit generality across tasks and graphs within a given domain. The acquired knowledge is then encoded in the weights of pre-trained GNNs. When it comes to downstream tasks, the initial weights can be efficiently refined through a lightweight fine-tuning step, leveraging a limited set of task-specific labels. However, as discussed in Sun et al. (2023a), the "pre-train and fine-tuning" paradigm is susceptible to the *negative transfer* problem.

Specifically, pre-trained GNN models focus on preserving the intrinsic graph properties, while fine-tuning seeks to optimize the weights on the downstream tasks, which may significantly differ from the pretext tasks employed in pre-training. For instance, consider the scenario where a GNN is pre-trained using link prediction objective Kipf & Welling (2016b), a prevalent pretext task that aims to bring the representations of adjacent nodes closer in latent space. Subsequently, fine-tuning is performed using the node classification objective. In such a case, the model might exhibit suboptimal performance or even break down, especially if the graph dataset is heterophilic, where adjacent nodes may have different labels.

Consequently, in an effort to narrow the gap between pre-training and downstream tasks, several exploratory graph prompting learning frameworks Liu et al. (2023); Sun et al. (2023a); Fang

et al. (2023) have been introduced. The concept of prompt tuning initially found application in the language domain Liu et al. (2022a); Li & Liang (2021a); Bhardwaj et al. (2022). In general, a piece of fixed or trainable prompt text is appended to the input text, aligning the downstream task with the text generation capabilities of pre-trained large language models (LLMs).

This approach not only preserves performance but also contributes to a reduction in training resource consumption. In the graph domain, prompt learning has recently demonstrated its potential as an alternative to fine-tuning, exemplified by methods such as GPPT Sun et al. (2022), GraphPrompt Liu et al. (2023), GPF Fang et al. (2023), and All-in-One Sun et al. (2023a). Similar to language prompts, these methods modify the original input graphs into prompted graphs which are further fed into frozen pre-trained graph models. The distinctions among these methods lie in the approach of inserting prompts into graphs and detailed training strategies. Nevertheless, the existing graph prompt learning approaches collectively operate under an assumption: *that the learned task-specific prompts perform well across all input instances within the task.* In other words, these prompts are considered static concerning the input, a limitation that we deem critical. We argue that the dependency of prompts on the input instance is an essential characteristic that aids in generalization over unseen samples, both in-domain and out-of-domain. Using two molecules from the BBBP dataset as an example, as shown in Figure 1, for molecule (a), it is acceptable to use one universal prompt vector for all the atoms (nodes). However, for molecule (b) with complex structures, it is evident that these highlighted atoms with red circles (i.e., S, C, and N), contain distinct features and should be prompted in different ways.

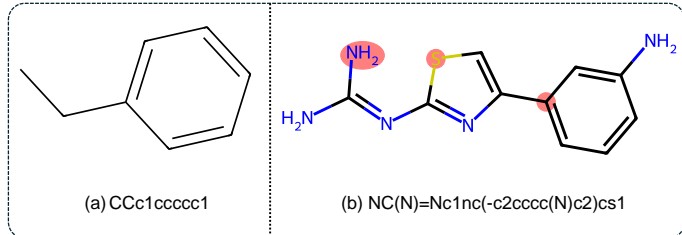

**Figure 1:** Two example molecules from the BBBP dataset. Molecule (a) with simple structures suffices with a universal prompt. However, molecule (b) with diverse atoms and intricate structures requires the use of instance-aware prompts.

To this end, our paper delves into the exploration of instance-aware prompt learning for the graph domain. This non-trivial research problem raises two questions: *(1) what model should we use to generate instance-aware prompts with additional use of a minimal number of parameters?* It is important to identify an effective and parameter-efficient method to transform the feature space into the prompt space, as the primary advantage of prompting lies in the minimization of trainable parameters. *(2) how can we ensure the instance-aware prompts are meaningful and distinctive as expected?* Employing parameterized methods for prompt generation runs the risk of converging to trivial solutions, where all prompts collapse into a singular solution. Consequently, guaranteeing the generation of diverse and meaningful prompts becomes a pivotal aspect of the entire pipeline.

In response to these challenges, we introduce a novel instance-aware graph prompt learning framework named IA-GPL designed to generate distinctive prompts for each instance by leveraging its individual information. Specifically, to tackle the first question, we feed the representations of the input instance into parameterized hypercomplex multiplication (PHM) layers Zhang et al. (2020) which transform the feature space into the prompt space with minimal parameters. To solve the second question, we resort to the vector quantization (VQ) GRAY (1998) technique. VQ discretizes the continuous space of intermediate prompts, mapping each prompt to a set of learnable codebook vectors. The mapped vectors after VQ then replace the original prompts and are incorporated into the original features. For the training of codebooks, the exponential moving average technique is utilized to prevent the model from converging to trivial solutions. To summarize, our main contributions are as follows:

- We propose IA-GPL, a novel instance-aware graph prompting framework. To the best of our knowledge, IA-GPL is the first graph prompting method capable of generating distinct prompts based on different instances within the dataset.

- In IA-GPL, we utilize a parameter-efficient bottleneck architecture for prompt generation followed by the vector quantization process via a set of codebook vectors and the exponential moving average technique to ensure effectiveness and stability.

- We conduct extensive experiments under different settings to evaluate the performance of IA-GPL. Our results demonstrate its superiority over other state-of-the-art competitors.

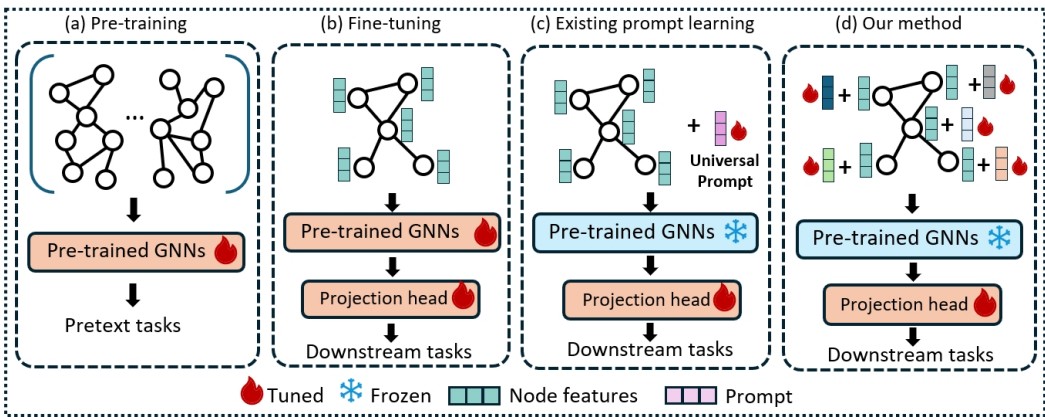

**Figure 2:** Comparison between different paradigms of graph representation learning.

## 2 RELATED WORK

**Due to space limit, an extended related work section is provided in Appendix B**. As mentioned in the introduction, to mitigate the reliance on graph labels, numerous GNN pre-training techniques have been proposed which can be implemented at node-level, edge-level, or graph-level Zhu et al. (2021); Jin et al. (2020); You et al. (2020a). Since fully fine-tuning the pre-trained GNN models may cause the negative transfer problem between pre-training tasks and downstream tasks, prompt-based methods Sun et al. (2023a); Liu et al. (2024), emerged to mitigate this issue. Existing graph prompt learning methods either integrate a prompt graph into the original graph Sun et al. (2023a) or inject a feature vector Fang et al. (2023) into the original features. However, these approaches do not differentiate between input instances and process them uniformly, which could be sub-optimal in certain scenarios. **Note that a concurrent work GPF-plus does incorporate different prompts for different nodes using the attention mechanism.** However, we believe that our method has several advantages over GPF-plus: (1) our method includes a lightweight down- and up-sample projector model that transforms the node hidden representations to another prompt vector space, while GPF-plus directly computes attention in the original feature space and then averages the weighted candidate prompts. An additional alignment between these two spaces is beneficial for the disentanglement of distinct information. (2) instead of using original node features to compute similarities, we use node features after the frozen GNN, which contain rich neighbor-aware information, further aiding in the prompt generation process.

Thus, in this work, we propose IA-GPL, a novel methodology designed to address the aforementioned issue by generating instance-aware prompts using the distinctive features in individual instances.

## 3 PRELIMINARIES

**Graphs.** Let $G = (V, E, \mathbf{X}, \mathbf{A})$ represent an undirected and unweighted graph, where $V$ is the set of nodes and $E$ is the set of edges. $\mathbf{X} \in \mathbb{R}^{|V| \times d}$ is the node feature matrix where the $i$-th row $\mathbf{x}_i$ is the $d$-dimensional feature vector of node $v_i \in V$. $\mathbf{A} \in \mathbb{R}^{|V| \times |V|}$ denotes the binary adjacent matrix with $\mathbf{A}_{i,j} = 1$ if $e_{i,j} \in E$ and $A_{i,j} = 0$ otherwise. $\mathcal{N}(v)$ denotes the neighboring set of node $v$.

**Graph Neural Networks.** Generally, GNNs with a message-passing mechanism can be divided into two steps. First, the representation of each node is updated by aggregating messages from its local neighboring nodes. Second, the aggregated messages are combined with the node's own representation. Given a node $v$, these two steps are formulated as:

$$m_v^{(l)} = \text{AGGREGATE}^{(l)}\{h_v^{(l-1)}, \forall u \in \mathcal{N}(v)\}, \tag{1}$$

$$h_v^{(l)} = \text{COMBINE}^{(l)}\{h_v^{(l-1)}, m_v^{(l)}\}, \tag{2}$$

where $m_v^{(l)}$ and $h_v^{(l)}$ denote the message vector and representation of node $v$ in the $l$-th layer, respectively. In the first layer, $h_v^0$ is initialized as the node features $\mathbf{X}$ and the output of the last layer $h_v^l$ can be used in downstream tasks.

**GNN Pre-training and Fine-tuning.** Given a pre-trained GNN model $f_\theta(\cdot)$ parameterized by $\theta$, a learnable projection head parameterized by $\phi$ and a downstream graph dataset $\mathcal{G} = \{(G_1, y_1), (G_2, y_2), \cdots, (G_n, y_n)\}$, we update the parameters of the pre-trained model and the projection head to maximize the likelihood of predicting the correct labels $Y$ of the dataset $\mathcal{G}$:

$$\max_{\theta, \phi} P_{\theta, \phi}(Y | \mathcal{G}). \tag{3}$$

Specifically, if we only update the parameters of the projection head, it is referred to as linear probing:

$$\max_{\phi} P_{\theta, \phi}(Y | \mathcal{G}). \tag{4}$$

**Graph Prompt Learning.** Compared with fine-tuning, prompt learning introduces a prompt generation model that aims to obtain a prompted graph $g_\Phi : G \to G$ parameterized by $\Phi$. This model transforms an input graph $G$ to a prompted graph $g_\Phi(G)$ which replaces the original graph and is fed into the pre-trained graph model as normal. The pre-trained graph model is fixed while only the parameters of the projection head and the prompt generation model are updated:

$$\max_{\phi, \Phi} P_{\theta, \phi}(Y | g_\Phi(\mathcal{G})\}). \tag{5}$$

A visual comparison of these methods is presented in Figure 2. Note that, unlike other prevailing prompting frameworks that employ a universal prompt, our model integrates instance-aware prompts.

## 4 METHODOLOGY

In this section, we introduce the proposed framework of IA-GPL, as depicted in Figure 3. Firstly, we present a conceptual overview of the entire framework in section 4.1. Subsequently, we delve into the key components of IA-GPL - a lightweight bottleneck architecture consisting of PHM layers in section 4.2, a prompt quantization process via a set of codebook vectors in section 4.3, and model optimization with the exponential moving average technique in section 4.4.

### 4.1 NAIVE APPROACH

To generate prompts associated with input instances, the first step involves obtaining specific representations of these instances. So naturally we employ the pre-trained graph model $f_\theta(\cdot)$ as an encoder to generate the hidden embeddings:

$$\mathbf{H} = f_\theta(G), \quad z = \textsc{ReadOut}(\mathbf{H}), \tag{6}$$

where $G = (\mathbf{X}, \mathbf{A})$ is the input graph, $\mathbf{H} \in \mathbb{R}^{|V| \times d}$ is the obtained node representations and $z \in \mathbb{R}^d$ is the graph representation after ReadOut operation.

In IA-GPL, we consider **node-level instance-aware prompts** which means we generate different prompts for each node in the graph, as we unify different tasks into a general graph-level task following Sun et al. (2023a); Liu et al. (2024). Thus, after getting the node representations $\mathbf{H}$, we employ an efficient bottleneck multi-layer perceptron architecture as the prompt generation model to transform them into the prompt space. Specifically, we first project $\mathbf{H} \in \mathbb{R}^{|V| \times d}$ from $d$ to $d'$ dimensions ($d' < d$) followed by a nonlinear function. Then it is projected back to $d$ dimensions to get instance-aware prompts $\mathbf{P} \in \mathbb{R}^{|V| \times d}$, matching the same shape as $\mathbf{X}$ so that they can be added back to the original node features. Mathematically it can be formulated as:

$$g_\Phi(\cdot) = \textsc{UpProject}(\textsc{ReLU}(\textsc{DownProject}(\cdot))), \tag{7}$$

$$\mathbf{P} = g_\Phi(\mathbf{H}), \quad \mathbf{X_P} = \mathbf{X} + \mathbf{P}, \tag{8}$$

where $g_\Phi(\cdot)$ represents the prompt generation model, $\mathbf{X}$ is the original node features while $\mathbf{X_P}$ is the prompted node features which contain instance-dependent information. By far, we have established a general yet naive instance-aware prompt learning framework by replacing $G = (\mathbf{X}, \mathbf{A})$ with $G_p = (\mathbf{X_P}, \mathbf{A})$, and train the prompt generation model and the projection head using the back-propagation algorithm. To expand on this simple concept, the following sections will elaborate on the details of the lightweight prompt generation model and the optimization process.

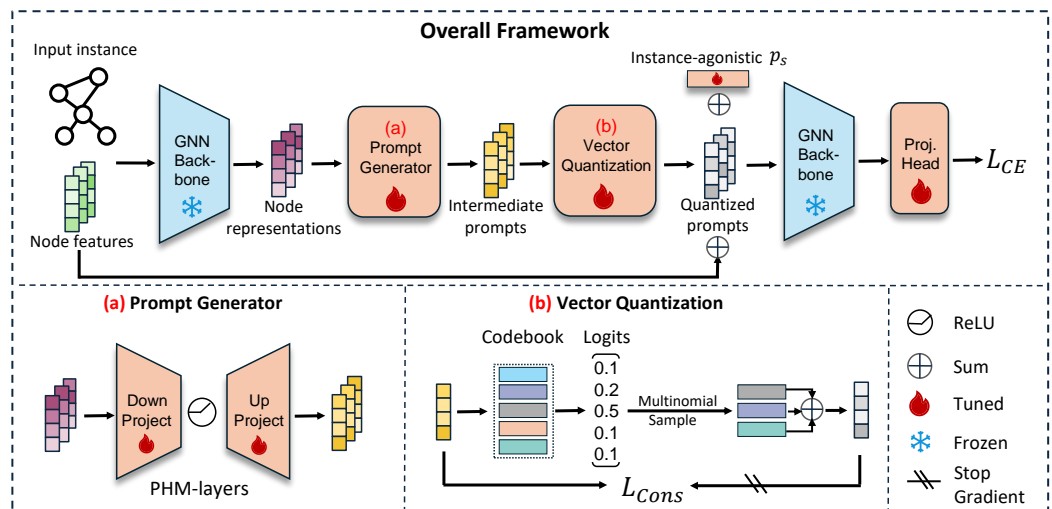

Figure 3: Overall Framework of IA-GPL.

## 4.2 Lightweight Bottleneck Architecture

The prompt generation model which transforms $\mathbf{H}$ from the feature space into the prompt space consists of a down-sample projector and an up-sample projector. Instead of the common option, FCN layers, we adopt PHM layers Zhang et al. (2020) which are more parameter-efficient.

**FCN layers.** One straightforward approach for implementing these two projectors is through fully connected layers (FCNs) which transform an input $\mathbf{x} \in \mathbb{R}^d$ into an output $\mathbf{y} \in \mathbb{R}^k$ by:

$$\mathbf{y} = \text{FC}(\mathbf{x}) = \mathbf{W}\mathbf{x} + \mathbf{b}, \tag{9}$$

where the weight matrix $\mathbf{W} \in \mathbb{R}^{k \times d}$ and the bias vector $\mathbf{b} \in \mathbb{R}^k$ are trainable parameters. We can control the number of parameters by controlling the hidden dimension $d'$, but it is a trade-off between performance and efficiency. In other words, it contradicts the original objective of prompt learning, which aims to reduce the number of trainable parameters, if we set $d'$ large to maintain performance.

**PHM layers.** To mitigate this problem, we turn to parameterized hyper-complex multiplication (PHM) layers as a compromise solution which can also be written in the similar way:

$$\mathbf{y} = \text{PHM}(\mathbf{x}) = \mathbf{M}\mathbf{x} + \mathbf{b}, \tag{10}$$

where the replaced parameter matrix $\mathbf{M} \in \mathbb{R}^{k \times d}$ is constructed by a sum of Kronecker products of several small matrices. The Kronecker product $\mathbf{X} \otimes \mathbf{Y}$ is defined as a block matrix:

$$\mathbf{X} \otimes \mathbf{Y} = \begin{bmatrix} x_{11}Y & \dots & x_{1n}Y \\ \vdots & \ddots & \vdots \\ x_{m1}Y & \dots & x_{mn}Y \end{bmatrix} \in \mathbb{R}^{mp \times nq}, \tag{11}$$

where $x_{ij}$ is the element of $\mathbf{X} \in \mathbb{R}^{m \times n}$ at its $i$-th row and $j$-th column and $\mathbf{Y} \in \mathbb{R}^{p \times q}$. Given a user-defined hyperparameter $n \in \mathbb{Z}_{>0}$, for $i = 1, 2, \dots, n$, let each parameter matrix be denoted as $\mathbf{A}_i \in \mathbb{R}^{n \times n}$ and $\mathbf{S}_i \in \mathbb{R}^{\frac{k}{n} \times \frac{d}{n}}$. Finally the parameter $\mathbf{M}$ is calculated by:

$$\mathbf{M} = \sum_{i=1}^{n} \mathbf{A}_i \otimes \mathbf{S}_i. \tag{12}$$

By replacing $\mathbf{W}$ with $\mathbf{M}$, the number of trainable parameters is reduced to $n \times (n \times n + \frac{m}{n} \times \frac{d}{n}) = n^3 + \frac{m \times d}{n}$. As $n$ is usually set as a small number (e.g., 2, 4, 8), the parameter size of a PHM layer is approximately $\frac{1}{n}$ of that of an FCN layer.

In the case of our approach, after we have node representations $\mathbf{H}$ through the pre-trained graph model, we feed them into the parameter-efficient PHM layers instead of standard FCN layers to generate instance-aware intermediate prompts $\mathbf{P}_c$.

### 4.3 PROMPT QUANTIZATION

Directly using $\mathbf{P}_c$ as prompts suffers from the high variance problem since there is no explicit constraint in the PHM layers, thus Vector Quantization (VQ) GRAY (1998) is utilized to discrete the intermediate prompt space $\mathbf{P}_c$ to $\mathbf{P}_q$. VQ is a natural and widely used method in signal processing and data compression that represents a set of vectors by a smaller set of representative vectors. This approach not only helps reduce the high variance caused by the PHM layers but also clusters similar hidden prompt representations together to provide the beneficial property of clustering.

Specifically, we maintain K trainable codebook vectors $\mathbf{E} = (\mathbf{e}_1, \mathbf{e}_2, \dots, \mathbf{e}_k) \in \mathbb{R}^{k \times d}$ shared across all the intermediate prompts $\mathbf{P}_c$. For every prompt $\mathbf{p}_c \in \mathbf{P}_c$, we sample M codebook vectors from $\mathbf{E}$ corresponding to $\mathbf{p}_c$ to obtain the quantized $\mathbf{p}_q$. Please note that the quantization process for each intermediate prompt $\mathbf{p}_c$ operates independently of other prompts. In detail, we first compute the squared Euclidean distance $d_c^i$ between the prompt $\mathbf{p}_c$ and every codebook vector $\mathbf{e}_i$, and the corresponding sampling logits $l_c^i$:

$$d_c^i = \|\mathbf{p}_c - \mathbf{e}_i\|_2^2, \quad l_c^i = -\frac{1}{\tau} d_c^i, \tag{13}$$

where $\tau$ is a temperature hyperparameter used to control the diversity of the sampling process. Then we sample $M$ latent codebook vectors with replacement for prompt $\mathbf{p}_c$ from a Multinomial distribution over the logits $l_c^i$:

$$z_c^1, z_c^2, \dots, z_c^M \sim \text{Multinomial}(l_c^1, l_c^2, \dots l_c^K). \tag{14}$$

Finally, the quantized prompt $p_q$ can be computed by averaging over the M sampled vectors:

$$\mathbf{p}_q = \frac{1}{M} \sum_{i=1}^{M} \mathbf{e}_{z_c^i}. \tag{15}$$

After the VQ process, we ensure that for semantically similar instances, the quantized prompts will also have similar representations by treating VQ as a clustering mechanism. In the meanwhile, the limited set of learnable codebook vectors explicitly constrains the information capacity of prompt representations $\mathbf{p}_q$, reducing the variance w.r.t. the output of PHM layers, $\mathbf{p}_c$.

Notably, we also introduce a learnable instance-agonist prompt $\mathbf{p}_s$ which is shared across all instances and incorporated into each quantized prompt $\mathbf{p}_q$ to have the final prompts $\mathbf{p}_f$:

$$\mathbf{p}_f = \mathbf{p}_q + \beta \mathbf{p}_s, \tag{16}$$

where $\beta$ is a balancing hyperparameter. This allows us to effectively fuse the learned information from the input-dependent aspects captured by $\mathbf{p}_q$ with the input-agnostic prompt $\mathbf{p}_s$.

In summary, given the high-variance prompts $\mathbf{p}_c$ after PHM layers in the last section, the application of VQ discretizes them into robust quantized prompts $\mathbf{p}_q$ that encapsulate intrinsic clustering property.

### 4.4 MODEL OPTIMIZATION

The PHM layers $\text{PHM}(\cdot)$, instance-independent static prompt $\mathbf{p}_s$, codebook vectors $\mathbf{E}$ and the projection head $\phi$ comprise the trainable parameters while we freeze the pre-trained GNN backbone $f_\theta(\cdot)$. The loss function is defined as:

$$\mathcal{L} = \mathcal{L}_{CE}(Y, Y_p) + \lambda \sum_{i=1}^{n} \|\mathbf{p}_{q_i} - \mathbf{p}_{c_i}\|_2^2, \tag{17}$$

which consists of two parts: *(1) Cross-entropy loss* between the ground truth $Y$ and the predicted labels $Y_p$ with prompted graphs as input. *(2) Consistency loss* that encourages the quantized prompts $\mathbf{p}_q$ to be consistent with the intermediate prompts $\mathbf{p}_c$ after PHM layers for all the $n$ instances (nodes) in the graph. These two terms collectively aim to preserve performance while minimizing information loss during the vector quantization process. $\lambda$ is a hyperparameter used to balance the two loss terms which is set to 0.01.

However, a potential limitation of directly training the model using back-propagation (BP) is representation collapse where all prompts become a constant embedding that disregards the input, causing

our model to degrade to GPF Fang et al. (2023). Thus to solve this problem, we still use the standard BP algorithm to update the PHM layers $\mathrm{PHM}(\cdot)$, instance-independent static prompt $\mathbf{p}_s$ and the projection head $\phi$ but adopt the exponential moving average (EMA) strategy to update the codebook vectors $\mathbf{E}$ following Angelidis et al. (2021); Roy et al. (2018). Specifically, for each batch in the training process, we perform the following two steps:

**Step 1:** Count the number of times the $j$-th codebook vector is sampled and update the count $c_j$:

$$c_{j_{(new)}} = \alpha \cdot c_{j_{(old)}} + (1 - \alpha) \cdot \sum_{i=1}^{n} \sum_{k=1}^{m} \mathbb{I}[\mathbf{e}_{z_i^k} = \mathbf{e}_j]. \tag{18}$$

**Step 2:** Update the embedding of $j$-th codebook vector $\mathbf{e}_j$ by calculating the mean of PHM layer outputs for which that codebook vector was sampled during Multinomial sampling:

$$\mathbf{e}_{j_{(new)}} = \alpha \cdot \mathbf{e}_{j_{(old)}} + (1 - \alpha) \cdot \sum_{i=1}^{n} \sum_{k=1}^{m} \frac{\mathbb{I}[\mathbf{e}_{z_i^k} = \mathbf{e}_j]\mathbf{p}_i^c}{c_j}, \tag{19}$$

where $n$ and $m$ stand for batch size and sample number, $\alpha$ is a hyperparameter set to 0.99 and $\mathbb{I}[\cdot]$ is the indicator function. By incorporating the EMA mechanism, we can avoid the representation collapse problem and also obtain a more stable training process than gradient-based methods.

## 5 EXPERIMENTS

### 5.1 EXPERIMENTAL SETUP

**Tasks and datasets.** We evaluate IA-GPL using both node-level and graph-level tasks. Following Sun et al. (2023a); Liu et al. (2024), we unify these tasks into a general graph-level task by generating local subgraphs for the nodes of interest. For graph-level tasks, we use eight molecular datasets from MoleculeNet Wu et al. (2018). For node-level tasks, we use three citation datasets from Yang et al. (2016). These datasets vary in *size, labels, and domains*, serving as a comprehensive benchmark for our evaluations. A comprehensive description of these datasets can be found in Appendix C.

**Baselines.** To evaluate the effectiveness of IA-GPL, we compare it with state-of-the-art approaches across three primary categories. *(1) Supervised learning:* we employ GCN Kipf & Welling (2016a), GraphSAGE Hamilton et al. (2017) and GIN Xu et al. (2018). The base models and the projection head are all trained end-to-end from scratch. *(2) Pre-training and fine-tuning:* The base model is pre-trained using edge prediction Jin et al. (2020) for molecular datasets and graph contrastive learning You et al. (2020a) for citation datasets. For the complete fine-tuning (FT), the pre-trained model is fine-tuned along with the projection head. For linear probing (LP), we freeze the pre-trained model and exclusively train the projection head. *(3) Prompt learning:* All in One Sun et al. (2023a), GPF Fang et al. (2023) and GPF-plus Fang et al. (2023) are included. They all freeze the pre-trained base model while training the projection head and their respective prompt generation models.

**Settings and implementations.** To evaluate the performance of IA-GPL in both in-domain and out-of-domain scenarios, we split the molecular datasets in two distinct manners: *random split* and *scaffold split*. Scaffold split is based on the scaffold of the molecules so that the train/val/test set is more structurally different, making it appropriate for evaluating the model's generalization ability. In contrast, the random split is used to assess the model's in-domain prediction ability. We test IA-GPL using 5 different pre-training strategies: edge prediction Jin et al. (2020) (denoted as EdgePred), Deep Graph Infomax Veličković et al. (2018) (denoted as InfoMax), Attribute Masking Hu et al. (2020a) (Denoted as AttrMasking), Context Prediction Hu et al. (2020b) (Denoted as ContextPred) and Graph Contrastive Learning You et al. (2020b) (Denoted as GCL) methods to demonstrate our model's robustness. We report results in both full-shot and few-shot settings, utilizing the ROC-AUC score as the metric. The few-shot setting is tested because prompt learning with fewer parameters is naturally less susceptible to the risk of overfitting when given limited supervision. We perform five rounds of experiments and report the mean and standard deviation. GCN is adopted as our backbone model. For the baselines, based on the authors' code and default settings, we further tune their hyperparameters to optimize their performance. Additional implementation details are provided in Appendix E. The anonymous source code is publicly available at https://anonymous.4open.science/r/IA-GPL-ICLR25.

Table 1: 50-shot ROC-AUC (%) performance comparison on molecular prediction benchmarks using **random** split. **Bold numbers** represent the best results in the graph prompting field (shaded region) to which our method belongs. Underlined numbers represent the best results achieved by other methods.

| Tuning Strategies | Methods | BBBP | Tox21 | ToxCast | SIDER | ClinTox | BACE | HIV | MUV | Avg. |
|---|---|---|---|---|---|---|---|---|---|---|
| Supervised | GIN | 80.20±1.70 | 64.55±1.14 | 53.77±2.32 | 52.11±1.51 | 52.68±4.62 | 69.14±1.17 | 62.87±2.52 | 49.17±5.92 | 60.56 |
| | GCN | 83.97±0.86 | 64.65±0.73 | 51.35±1.43 | 48.54±0.75 | 59.22±2.64 | 71.91±1.74 | 59.91±1.06 | 50.85±4.02 | 61.3 |
| | GraphSAGE | 80.72±1.37 | 63.91±1.08 | 52.09±0.43 | 49.14±1.19 | 59.57±2.40 | 71.33±0.97 | 61.06±1.34 | 53.08±5.38 | 61.36 |
| Pre-training+ Fine-tuning | Linear Probing | 79.67±1.31 | 69.99±0.27 | 61.74±0.48 | 52.61±0.39 | 70.33±3.76 | 76.17±0.77 | 65.04±1.49 | 59.12±1.33 | 66.83 |
| | Fine Tuning | 88.30±3.09 | 69.25±0.73 | 60.42±0.55 | 52.32±0.10 | 72.09±2.74 | 74.97±0.62 | 64.12±0.90 | 54.17±2.11 | 66.95 |
| Prompt Learning | All in One | 49.49±5.32 | 52.45±2.23 | 50.33±5.05 | 51.24±2.06 | 57.65±11.11 | 53.22±7.14 | 46.31±7.50 | - | 51.52 |
| | GPF | 82.86±1.98 | 69.56±2.50 | 61.11±0.43 | 52.24±0.16 | 73.31±4.08 | 76.54±1.76 | 63.21±0.53 | 59.14±1.02 | 67.24 |
| | GPF-plus | 83.08±1.57 | 71.31±0.80 | 60.85±1.69 | 52.44±0.83 | 73.85±2.15 | 76.02±0.99 | 64.49±1.19 | **59.93±0.83** | 67.74 |
| | IA-GPL | **85.62±0.52** | **72.55±0.40** | **61.63±0.40** | **52.85±0.84** | **74.50±0.76** | **76.64±0.83** | **64.60±0.95** | 59.32±1.13 | **68.46** |

Table 2: 50-shot ROC-AUC (%) performance comparison on molecular prediction benchmarks using **scaffold** split. **Bold numbers** represent the best results in the graph prompting field (shaded region) to which our method belongs. Underlined numbers represent the best results achieved by other methods.

| Tuning Strategies | Methods | BBBP | Tox21 | ToxCast | SIDER | ClinTox | BACE | HIV | MUV | Avg. |
|---|---|---|---|---|---|---|---|---|---|---|
| Supervised | GIN | 56.92±2.54 | 46.83±1.51 | 52.50±0.68 | 48.85±2.16 | 50.00±7.53 | 51.08±2.14 | 68.09±3.89 | 49.11±2.45 | 52.92 |
| | GCN | 57.05±5.50 | 47.40±3.56 | 49.67±0.61 | 49.93±1.06 | 59.84±5.54 | 61.84±2.12 | 62.82±2.56 | 42.44±3.40 | 53.87 |
| | GraphSAGE | 59.13±7.28 | 48.42±3.01 | 51.90±1.43 | 49.60±1.92 | 40.53±4.25 | 59.28±1.60 | 64.28±1.09 | 49.11±2.90 | 52.78 |
| Pre-training+ Fine-tuning | Linear Probing | 52.54±5.77 | 64.40±0.42 | 57.46±0.33 | 50.76±0.74 | 62.54±4.26 | 59.75±4.23 | 61.89±4.10 | 63.07±3.09 | 59.05 |
| | Fine-tuning | 48.88±0.68 | 60.95±1.46 | 55.73±0.43 | 51.30±2.21 | 57.78±4.03 | 61.27±6.10 | 62.20±4.95 | 64.75±2.03 | 57.85 |
| Prompt Learning | All in One | 53.46±7.98 | 56.19±4.96 | 55.35±2.12 | 51.51±2.82 | 48.91±16.03 | 52.90±7.90 | 39.89±6.09 | - | 51.17 |
| | GPF | 52.13±1.21 | 63.48±0.41 | 57.60±0.19 | 51.07±1.08 | **65.18±1.76** | 58.78±5.04 | 65.59±2.31 | 66.94±3.91 | 60.09 |
| | GPF-plus | 54.73±5.20 | 63.29±0.55 | 57.19±0.67 | 50.31±1.60 | 64.14±2.95 | 55.87±7.40 | 61.4±4.30 | 67.11±2.09 | 59.25 |
| | IA-GPL | **56.54±2.35** | **64.14±0.44** | **58.11±0.38** | **53.18±1.18** | 63.28±3.52 | **61.95±4.00** | **66.52±2.10** | **69.03±3.02** | **61.59** |

## 5.2 PERFORMANCE EVALUATION

Due to the page limit, we present the experimental results of 50-shot random split and scaffold split settings on molecular datasets using edge prediction pre-training strategy in Table 1 and Table 2. **The results of full-shot learning, node-level tasks, larger graph datasets and more pre-training strategies results are provided in Appendix D.**

**In-domain performance.** Table 1 illustrates the results for 50-shot graph classification under the in-domain setting (random split). We have the following observations: (1) Compared to the pre-training and fine-tuning approach, IA-GPL achieves competitive results despite employing a significantly lower number of trainable parameters. This underscores the key advantage of prompt learning, particularly when confronted with limited supervision. (2) While fine-tuning occasionally outperforms IA-GPF on certain datasets, IA-GPF consistently surpasses other graph prompt learning methods as shown in the shaded area, highlighting the significance of employing instance-aware prompts. (3) Unexpectedly, the All-in-One approach lags behind other prompting methods, exhibiting the highest variance. This discrepancy may be attributed to an unstable training process.

**Out-of-domain performance.** Table 2 illustrates the results for 50-shot graph classification under the out-of-domain setting (scaffold split). We have the following observations: (1) Overall, IA-GPL attains optimal results across these eight datasets, underscoring its efficacy even when confronting the out-of-distribution (OOD) challenge. We attribute this success to the vector quantization process, which captures the clustering property of molecules. The disentangled clustering information can enhance performance in the presence of OOD samples by facilitating the transfer of learned knowledge. (2) Across different datasets, the performance trends of supervised learning, pre-training and fine-tuning, and prompt learning paradigms vary a lot. For instance, training GCN, GIN, or GraphSAGE in an end-to-end manner yields the highest performance in the BBBP dataset, whereas it performs less effectively in other datasets such as Tox21 and SIDER. These fluctuations in performance may be attributed to the distinctive intrinsic properties characterizing each dataset.

When considering the broader context, several key observations emerge: (1) Comparing linear probing (LP) and fine-tuning (FT), the performance trend differs between random and scaffold split. For random split, FT outperforms LP, whereas the reverse is observed for scaffold split. This observation

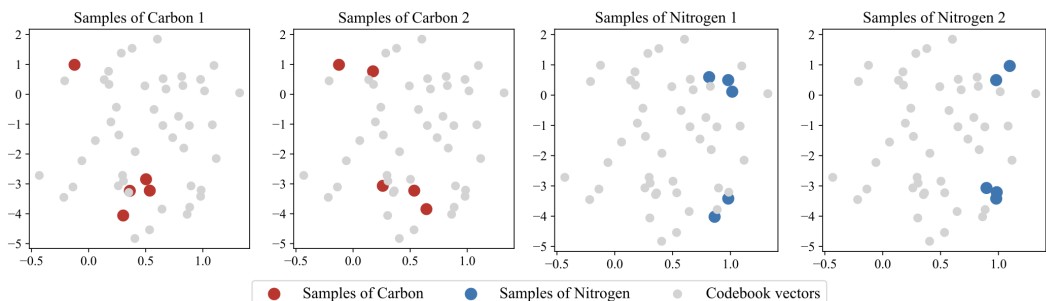

**Figure 4:** Codebook visualization.

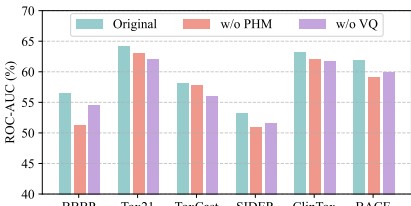

| Models | #Tuning parameters | Relative ratio | Training time per epoch | GPU memory consumption |
|---|---|---|---|---|
| Fine-tuning | 1.86M | 100% | ∼0.68s | ∼796MB |
| GPF | 0.3K | 0.02% | ∼0.81s | ∼768MB |
| GPF-plus | 3-12K | 0.16-0.65% | ∼0.82s | ∼740MB |
| All-in-One | 3K | 0.16% | - | - |
| IA-GPL (Ours) | 20K | 1.08% | ∼0.86s | ∼780MB |

**Figure 5:** Ablation study.          **Figure 6:** Model efficiency analysis.

confirms the negative transfer drawback associated with the "pre-training and fine-tuning" paradigm: the existence of a gap between pretext tasks and downstream tasks leads to suboptimal performance. (2) The performance gain achieved by IA-GPL is more pronounced in the out-of-domain scenario, emphasizing the importance of vector quantization within our model. (3) The overall performance for in-domain classification remains significantly better than that for out-of-domain classification, underscoring the imperative to design effective methods to address the OOD problem.

## 5.3 MODEL ANALYSIS

**Codebook visualization.** We conduct a visualization and interpretability analysis on the learned codebook using a molecule from the BACE dataset with the SMILES string `O=C1NC(=NC(=C1)CCC)N` as an example. The model is configured to have 50 codebook vectors in the VQ space. For every node (atom), we sample 5 vectors using Equation 14, which are then averaged and used as quantized prompts. Figure 4 presents the t-SNE Van der Maaten & Hinton (2008) plots of the samples of two carbon atoms and two nitrogen atoms in this molecule.

Two characteristics of the learned codebooks are observed: (1) Samples corresponding to different atoms manifest substantial distinctions (i.e., the regions of samples in the plot). However, samples corresponding to the same atoms tend to exhibit in proximate regions in the codebook vector space. This observation affirms that IA-GPL effectively generates instance-aware prompts. (2) Each atom's samples may also demonstrate a clustering property. This phenomenon may be attributed to the disentanglement of representations for individual instances within the prompt space, which potentially encompasses general information.

**Ablation study.** To assess the individual contributions of each component, we conduct an ablation study by comparing IA-GPL with two different variants: *(1) w/o VQ:* After getting $P_c$ through the PHM layers, we directly use it as the final prompts without the vector quantization process. *(2) w/o PHM:* We replace PHM layers in the prompt generation model with the standard MLP layers. Note that to maintain a fair comparison and ensure a roughly equivalent number of trainable parameters, we reduce the size of the hidden dimension to $\frac{1}{n}$ of that of PHM layers, as discussed in Section 4.2.

We conduct the ablation study under 50-shot learning with scaffold split and illustrate the results in Figure 5. We have the following observations: (1) Replacing PHM layers with MLP layers of the same parameter size adversely affects performance to varying degrees across these datasets. This result highlights the advantage of PHM layers over MLP layers when training resources are limited.

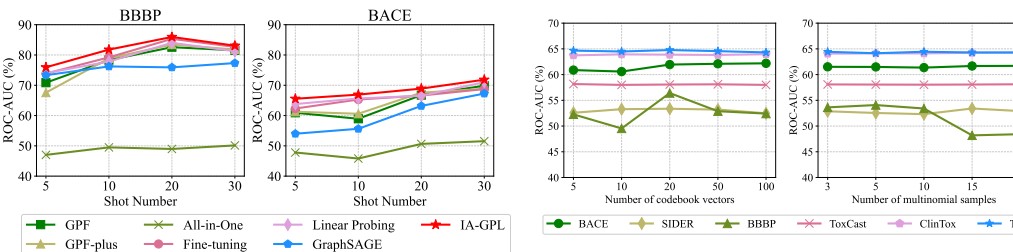

**Figure 7:** Impact of shot numbers.

**Figure 8:** Impact of VQ hyperparameters.

(2) Without the VQ process, the results drop as there is no constraint to prevent codebook vectors from collapsing which leads to inferior performance or an unstable training process.

**Efficiency analysis.** We analyze IA-GPL's *parameter efficiency* and *training efficiency* in Table 6.

In terms of parameter efficiency, we compute the number of tunable parameters for different strategies. (excluding the task-specific projection head). Specifically, fine-tuning demands the update of all parameters, making it the most time- and resource-consuming process. In the prompt learning domain, GPF is the most efficient since it requires only one universal prompt while GPF-plus incorporates multiple attentive prompts. All-in-One also utilizes more than one prompt node to construct a prompt graph. In our model, the parameter size is predominantly dominated by the prompt generation model (i.e., PHM layers), which aligns with the scale of other graph prompt learning methods and is significantly smaller than the fine-tuning approach.

In terms of training efficiency, we compute the training time per epoch and GPU memory consumption on the ToxCast dataset using a single Nvidia RTX 3090. We keep all hyper-parameters the same including batch size, dimensions, etc. All-in-One is omitted due to its unsatisfactory performance and unstable training process. Surprisingly, prompt-based methods are slower than traditional fine-tuning which may be due to additional procedures such as computing attention scores and sampling. Regarding GPU memory consumption, prompt-based methods occupy slightly less GPU space since they do not need to save the gradients and optimizer states for the frozen GNN backbone. But all of them are at the same level considering the dominant overhead GPU consumption.

**Impacts of the shot number.** We study the impact of the number of shots on the BBBP and BACE datasets in the few-shot random split setting. We vary the number of shots within the range of [5,10,20,30] and results are illustrated in Figure 7. In general, our method IA-GPL consistently surpasses or attains comparable results with other graph prompt learning frameworks in most cases especially when given very limited labeled data. As the number of shots increases, the overall performance increases while conventional supervised methods become more competitive.

**Impacts of the codebook hyperparameters.** We investigate the impact of the number of codebook vectors and the number of samples in the vector quantization process. Specifically, we vary the size of the codebook within the range of [5, 10, 20, 50, 100] and the sample size within [3, 5, 10, 15, 20], while keeping the remaining hyperparameters constant. Results are illustrated in Figure 8. We observe that for most of the datasets, our model achieves a relatively stable performance with respect to the hyperparameters, alleviating the need for meticulous and specific tuning.

## 6 CONCLUSIONS

In this paper, we introduce a novel prompting method on graphs named Instance-Aware Graph Prompt Learning (IA-GPL), which is designed to generate distinct and specific prompts for individual input instances within a downstream task. Specifically, we initially generate intermediate prompts corresponding to each instance using a parameter-efficient bottleneck architecture. Subsequently, we quantize these prompts with a set of trainable codebook vectors and employ the exponential moving average strategy to update the parameters which ensures a stable training process. Extensive experimental evaluations under full-shot and few-shot learning settings showcase the superior performance of IA-GPL in both in-domain and out-of-domain scenarios.

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

## A  BROADER IMPACTS AND LIMITATIONS

**Broader Impacts.**  Research that is focused on parameter-efficient fine-tuning methods (PEFT) including our approach, IA-GPL, can usually bring several broad positive societal impacts such as: *(1) Accessibility and Sustainability.* By reducing the computational, financial, and environmental resources needed for fine-tuning pre-trained models, PEFT methods make advanced AI technologies accessible to a wider range of individuals, organizations, and communities, including those with limited resources. *(2) Improved Quality of AI Applications.* IA-GPL also enhances numerous graph-related tasks and applications. By incorporating node-level instance-aware prompts, IA-GPL is particularly well-suited for complex and heterogeneous graphs, such as molecules and social networks. Consequently, IA-GPL can positively impact areas like drug discovery, protein structure prediction, fraud detection, and so on.

**Limitations.** *(1) Larger parameter size.* IA-GPL demands a larger parameter size, primarily due to the PHM layers when generating prompts. As discussed in section 5.3, compared to the huge pre-trained GIN model, the increase in the number of parameters remains acceptable considering the performance improvement achieved. However, it does introduce additional trainable parameters. *(2) More hyperparameters.* In addition to the standard hyperparameters such as the number of layers and embedding size, IA-GPL introduces new hyperparameters: the number of codebook vectors, the number of samples, and the temperature factor. However, as analyzed in Appendix 5.3, IA-GPL exhibits limited sensitivity to these hyperparameters across most datasets, obviating the need for meticulous and specific tuning.

## B  RELATED WORK

**Graph Representation Learning.** The objective of graph representation learning is to proficiently encode sparse high-dimensional graph-structured data into low-dimensional dense vectors. These vectors are subsequently employed in various downstream tasks, such as node/graph classification and link prediction. The methods span from classic graph embeddings Grover & Leskovec (2016) to recent graph neural networks Kipf & Welling (2016a); Veličković et al. (2017); Yun et al. (2019a) with the remarkable success of deep learning. GNNs, which derive effective node representations by recursively aggregating information from neighbor nodes, have emerged as a predominant standard for graph representation learning. GNNs find applications in diverse domains, such as social network analysis Guo & Wang (2020); Liu et al. (2021c), bioinformatics Mercado et al. (2021); Guo et al. (2021), recommendation systems Fan et al. (2019); Tian et al. (2022), and fraud detection Dou et al. (2020); Liu et al. (2021b). This is attributed to the fact that many real-world datasets inherently possess a graph structure, making GNNs well-suited for effectively modeling and extracting meaningful representations from such data. We refer the readers to a comprehensive survey Ju et al. (2023) for details.

**GNNs Pre-training.** Supervised learning methods applied to graphs heavily depend on graph labels, which may not always be adequate in real-world scenarios. To overcome this limitation, a pre-training and fine-tuning paradigm has been introduced. In this approach, GNNs are initially pre-trained to capture extensive knowledge from a substantial volume of labeled and unlabeled graph data. Subsequently, the implicit knowledge encoded in the model parameters is transferred to a new domain or task through the fine-tuning of partially pre-trained models. Existing effective pre-training strategies can be implemented at node-level like GCA Zhu et al. (2021), edge-level like edge prediction Jin et al. (2020), and graph-level such as GraphCL You et al. (2020a) and SimGRACE Xia et al. (2022). However, these methods overlook the gap that may exist between the pre-training phase and downstream objectives, limiting their overall generalization ability.

**Graph Prompt Learning.** Prompt Learning seeks to bridge the gap between pre-training and fine-tuning by formulating task-specific prompts that guide downstream tasks, with the pre-trained model parameters usually kept static during downstream applications. Many effective prompt methods were initially proposed in the natural language processing community, including hand-crafted prompts Gao et al. (2020); Schick & Schütze (2020) and continuous prompts Gu et al. (2021); Li & Liang (2021b); Liu et al. (2022b). Drawing inspiration from these works, several exploratory graph prompt learning methods, such as GPPT Sun et al. (2022), GraphPrompt Liu et al. (2023), GPF Fang et al. (2023) and All-in-One Sun et al. (2023a) have been proposed in the last two years. We recommend referring to the comprehensive survey provided in Sun et al. (2023b). These existing methods introduce virtual class-

prototype nodes or graphs with learnable links into the input graph or directly incorporate learnable embeddings into the representations, facilitating a closer alignment between downstream applications and the pretext tasks. However, all existing graph prompt tuning methods have predominantly concentrated on task-specific prompts, failing to generate instance-specific prompts which are critical since a universal prompt template may not effectively accommodate input nodes and graphs with significant diversity as shown in Figure 1. In this work, we introduce IA-GPL, a novel methodology designed to address the aforementioned issue by generating prompts that leverage the distinctive features in individual instances.

## C  ADDITIONAL DATASET DETAILS

We have two kinds of tasks and corresponding datasets: *graph-level tasks*: molecular datasets and *node-level tasks*: citation networks. The statistics of these datasets are illustrated in Table 3.

For molecular datasets, during the pre-training process, we sample 2 million unlabeled molecules from the ZINC15 Sterling & Irwin (2015) database, along with 256K labeled molecules from the preprocessed ChEMBL Mayr et al. (2018); Gaulton et al. (2011) dataset. For downstream tasks, we use the molecular datasets from MoleculeNet Wu et al. (2018) encompassing molecular graphs spanning the domains of physical chemistry, biophysics, and physiology. Specifically, they involve 8 molecular datasets: BBBP, Tox21, ToxCast, SIDER, Clintox, BACE, HIV and MUV. All datasets come with additional node and edge features introduced by open graph benchmarks Hu et al. (2020c).

For citation networks, we use 3 commonly used datasets: Cora, CiteSeer, and PubMed from Yang et al. (2016). Nodes represent documents and edges represent citation links. Each document (node) in the graph is described by a 0/1-valued word vector indicating the absence/presence of the corresponding word from the dictionary. During the pre-training phase, we use them without labels in a self-supervised learning approach. In the fine-tuning stage, we convert the node-level task to the graph-level task following Sun et al. (2023a) and process them in the same way as the molecular datasets.

Table 3: Statistics of the datasets.

| Tasks | Name | #graphs | #nodes | #edges | #features | #binary tasks/classes |
|-------|------|---------|--------|--------|-----------|----------------------|
| Graph-level | BBBP | 2,050 | $\sim$23.9 | $\sim$51.6 | 9 | 1 |
| | Tox21 | 7,831 | $\sim$18.6 | $\sim$38.6 | 9 | 12 |
| | ToxCast | 8,597 | $\sim$18.7 | $\sim$38.4 | 9 | 617 |
| | SIDER | 1,427 | $\sim$33.6 | $\sim$70.7 | 9 | 27 |
| | ClinTox | 1,484 | $\sim$26.1 | $\sim$55.5 | 9 | 2 |
| | BACE | 1,513 | $\sim$34.1 | $\sim$73.7 | 9 | 1 |
| | MUV | 93,087 | $\sim$24.2 | $\sim$52.6 | 9 | 17 |
| | HIV | 41,127 | $\sim$25.5 | $\sim$54.9 | 9 | 1 |
| Node-level | Cora | 1 | 2,708 | 10,556 | 1,433 | 7 |
| | CiteSeer | 1 | 3,327 | 9,104 | 3,703 | 6 |
| | PubMed | 1 | 19,717 | 88,648 | 500 | 3 |

## D  ADDITIONAL EXPERIMENTAL RESULTS

### D.1  RESULTS OF FULL-SHOT LEARNING

We present the experimental results using the full datasets to train the model in both scaffold split and random split scenarios in Table 4 and Table 5, respectively.

**In-domain performance.** Table 5 illustrates the results for full-shot graph classification under the in-domain setting (random split). We have the following observations: (1) Overall, fine-tuning exhibits superior performance across all methods including supervised schemes and prompt learning

Table 4: Full-shot ROC-AUC (%) performance comparison on molecular prediction benchmarks using scaffold split. **Bold numbers** represent the best results in the graph prompting field (shaded region) to which our method belongs. Underlined numbers represent the best results achieved by other methods.

| Tuning Strategies | Methods | BBBP | Tox21 | ToxCast | SIDER | ClinTox | BACE | HIV | MUV | Avg. |
|---|---|---|---|---|---|---|---|---|---|---|
| Supervised | GIN | 67.30±2.80 | 74.23±0.65 | 62.22±1.31 | 57.43±1.24 | 48.83±3.03 | 72.78±2.48 | 75.82±2.89 | 74.79±1.37 | 66.68 |
| | GCN | 62.18±3.49 | 74.48±0.55 | 62.74±0.59 | 62.51±1.06 | 56.58±3.22 | 73.44±1.64 | 78.26±2.01 | 71.98±2.34 | 67.77 |
| | GraphSAGE | 67.91±2.58 | 74.14±0.55 | 63.79±0.70 | 62.80±1.15 | 58.04±5.68 | 69.27±2.91 | 75.77±3.09 | 71.90±1.43 | 67.95 |
| Pre-training+ Fine-tuning | Linear Probing | 69.45±0.58 | 79.55±0.12 | 65.41±0.41 | 66.39±0.79 | 67.41±1.77 | 83.10±0.44 | 76.87±1.98 | 80.42±1.03 | 73.57 |
| | Fine Tuning | 66.56±3.56 | 78.67±0.35 | 66.29±0.45 | 64.35±0.78 | 69.07±4.61 | 80.90±0.92 | 79.79±2.76 | 81.76±1.80 | 73.42 |
| Prompt Learning | GPPT | 64.13±0.14 | 66.41±0.04 | 60.34±0.14 | 54.86±0.25 | 59.81±0.46 | 70.85±1.42 | 60.54±0.54 | 63.05±0.34 | 62.49 |
| | GPPT (w/o ol) | **69.43±0.18** | 78.91±0.15 | 64.86±0.11 | 60.94±0.18 | 62.15±0.69 | 70.31±0.99 | 73.19±0.19 | 82.06±0.53 | 70.23 |
| | GraphPrompt | 69.29±0.19 | 68.09±0.19 | 60.54±0.21 | 58.71±0.13 | 55.37±0.57 | 67.70±1.26 | 59.31±0.93 | 62.35±0.14 | 62.67 |
| | All in One | 58.01±4.89 | 52.38±3.46 | 55.07±7.22 | 53.33±2.16 | 50.91±9.33 | 55.86±12.75 | 58.32±4.40 | - | 54.84 |
| | GPF | 68.87±0.57 | 79.93±0.08 | 65.63±0.41 | 65.93±0.64 | 66.40±2.77 | 80.37±4.07 | 75.20±1.30 | 80.87±1.76 | 73.47 |
| | GPF-plus | 68.16±0.78 | 79.59±0.09 | 65.22±0.32 | 66.08±0.85 | 71.23±3.01 | 82.15±1.64 | 76.99±2.01 | 81.93±1.68 | 73.91 |
| | IA-GPL | 69.25±0.06 | **80.28±0.20** | **65.87±0.64** | **66.62±1.23** | **71.96±0.41** | **83.38±0.94** | **78.86±1.38** | **83.26±1.77** | **74.93** |

frameworks which is not surprising. Given an ample amount of labeled training data, fine-tuning can effectively adapt the pre-trained model that already encapsulates intrinsic graph properties, thereby contributing to optimal performance. (2) IA-GPL consistently attains the highest results in the realm of graph prompt learning, demonstrating its exceptional performance in this category and the importance of instance-aware prompts.

Table 5: Full-shot ROC-AUC (%) performance comparison on molecular prediction benchmarks using random spilt. **Bold numbers** represent the best results in the graph prompting field (shaded region) to which our method belongs. Underlined numbers represent the best results achieved by other methods.

| Tuning Strategies | Methods | BBBP | Tox21 | ToxCast | SIDER | ClinTox | BACE | HIV | MUV | Avg. |
|---|---|---|---|---|---|---|---|---|---|---|
| Supervised | GIN | 93.09±0.94 | 82.47±0.68 | 70.71±0.45 | 57.76±1.42 | 75.61±3.57 | 87.90±1.49 | 81.96±1.90 | 80.57±2.02 | 78.76 |
| | GCN | 92.59±0.79 | 81.82±0.23 | 72.50±0.55 | 57.10±0.95 | 80.45±3.26 | 88.09±0.60 | 83.06±0.45 | 79.18±1.86 | 79.35 |
| | GraphSAGE | 91.98±0.49 | 82.52±0.32 | 72.55±0.42 | 56.65±1.18 | 80.57±2.02 | 88.05±1.90 | 83.43±1.34 | 79.61±2.93 | 79.42 |
| Pre-training+ Fine-tuning | Linear Probing | 88.21±0.05 | 82.86±0.12 | 74.55±0.25 | 61.16±0.54 | 85.51±1.09 | 89.73±0.52 | 85.42±0.68 | 89.53±0.42 | 82.12 |
| | Fine Tuning | 93.06±0.35 | 85.46±0.26 | 75.35±0.33 | 63.89±0.69 | 87.22±1.12 | 90.93±0.55 | 86.84±0.72 | 87.26±0.76 | 83.75 |
| Prompt Learning | All in One | 62.88±9.60 | 52.38±3.46 | 45.24±8.53 | 48.78±4.17 | 44.86±18.81 | 51.82±4.01 | 54.78±1.76 | - | 51.53 |
| | GPF | **92.71±0.38** | 83.53±0.35 | 73.53±0.35 | 61.96±1.08 | **90.65±0.53** | 86.83±0.26 | 85.63±0.39 | 90.29±0.14 | 83.08 |
| | GPF-plus | 89.91±0.22 | 83.04±0.70 | 74.24±0.36 | 62.50±1.38 | 88.72±0.64 | 88.56±0.52 | 85.26±0.81 | 91.13±0.16 | 83.67 |
| | IA-GPL | 91.77±0.40 | **84.15±0.29** | **75.64±0.44** | **62.61±0.73** | 87.27±0.97 | **90.14±0.14** | **86.02±0.90** | **91.57±0.19** | **85.90** |

**Out-of-domain performance.** Table 4 illustrates the results for full-shot graph classification under the out-of-domain setting (scaffold split). We have the following observations: (1) When addressing the out-of-domain problem, IA-GPL consistently showcases superior performance compared to other baselines, confirming the clustering benefit derived from the vector quantization process. (2) While supervised learning can yield acceptable results in the in-domain setting, it notably lags behind fine-tuning and prompt learning approaches when confronted with the out-of-domain challenge. This underscores the benefit in generalization gained from the graph pre-training phase when a substantial amount of labeled and unlabeled data are available to equip the pre-trained model with prior knowledge.

## D.2 RESULTS OF NODE-LEVEL TASKS

We present the experimental results using node-level datasets-Cora, CiteSeer and PubMed in Table 6. We unify the task into a general graph-level task by generating local subgraphs for the nodes of interest and use the 100-shot setting following Sun et al. (2023a). We observe that (1) IA-GPL achieves the best performance on all three datasets, demonstrating its capacity in node-level tasks. (2) Supervised learning outperforms the fine-tuning approach by a large margin, showcasing the implicit negative transfer problem.

Table 6: 100-shot test accuracy (%) performance on node-level citation network datasets. **Bold numbers** represent the best results in the graph prompting field (shaded region) to which our method belongs. Underlined numbers represent the best results achieved by other methods.

| Tuning Strategies | Methods | Cora | CiteSeer | PubMed | Avg. |
|---|---|---|---|---|---|
| Supervised | GCN | 78.06±1.37 | 82.11±1.02 | 74.33±1.44 | 78.17 |
| | GAT | 79.71±0.77 | 82.27±0.68 | 74.44±0.68 | 78.81 |
| | TransformerConv | 78.50±0.68 | 82.66±0.36 | 75.00±1.24 | 78.72 |
| Pre-training+ Fine-tuning | Linear Probing | 60.53±4.07 | 82.05±0.20 | 70.22±1.25 | 70.93 |
| | Fine Tuning | 55.16±3.87 | 80.33±0.40 | 60.11±0.10 | 65.20 |
| Prompt Learning | All in One | 63.96±7.23 | 80.38±0.20 | 58.33±1.44 | 67.56 |
| | GPF | 70.13±1.58 | 77.67±1.24 | 58.67±1.58 | 68.82 |
| | GPF-plus | 71.43±1.04 | 78.67±0.92 | 61.33±1.29 | 70.48 |
| | IA-GPL | **71.51±0.97** | **81.33±1.29** | **63.33±0.67** | 72.06 |

Table 7: 50-shot and full-shot performance comparison on PPI dataset.

| Setting | GraphSAGE | GCN | GIN | Linear Probing | Fine Tuning | All-in-One | GPF | GPF-plus | IA-GPL |
|---|---|---|---|---|---|---|---|---|---|
| 50-shot | 37.20 | 40.75 | 39.56 | 49.70 | 46.23 | 42.90 | 50.64 | 52.56 | **53.13** |
| full-shot | 77.43 | 79.90 | 78.86 | 70.94 | 72.41 | 48.67 | 75.43 | 75.06 | **77.70** |

## D.3 RESULTS OF MORE PRE-TRAINING STRATEGIES

Besides the edge prediction Jin et al. (2020) pre-training strategies, we also use Deep Graph Infomax Veličković et al. (2018) (denoted as InfoMax), Attribute Masking Hu et al. (2020a) (Denoted as AttrMasking), Context Prediction Hu et al. (2020b) (Denoted as ContextPred) and Graph Contrastive Learning You et al. (2020b) (Denoted as GCL) methods to compare with IA-GPL to demonstrate our model's robustness. Note that we test under the full-shot scaffold split setting. Results are illustrated in Table 8. We observe that IA-GPL achieves state-of-the-art results in 27 out of 32 cases within the graph prompt learning area.

## D.4 RESULTS OF LARGER GRAPH DATASETS

Beyond the results of the previous molecular datasets, here we show the performance comparison of a relatively large biological dataset, PPI dataset which has 88k graphs and 40 classes. We tested it using the edge prediction pre-training strategy under both few-shot and full-shot settings. Results are illustrated in Table 7.

## E ADDITIONAL IMPLEMENTATION DETAILS

Table 9 presents the hyperparameter settings used during the adaptation stage of pre-trained GNN models on downstream tasks in IA-GPL. For molecular datasets, we adopt the widely used 5-layer GIN Xu et al. (2018) as the underlying architecture for our models. For citation networks, we adopt 2-layer Graph Transformers Yun et al. (2019b) as the underlying architecture. Grid search is used to find the best set of hyperparameters. You can also visit our code repository to obtain the specific commands for reproducing the experimental results. All the experiments are conducted using NVIDIA V100 graphic cards with 32 GB of memory and PyTorch framework. For the details, please visit our code repository.

Table 8: Full-shot ROC-AUC (%) performance comparison on molecular prediction benchmarks with Deep Graph Infomax, Attribute Masking, ContextPred and GCL as pre-training methods. **Bold numbers** represent the best results in the graph prompting field to which our method belongs. Underlined numbers represent the best results achieved by other methods.

| Pre-training Strategies | Tuning Strategies | Methods | BBBP | Tox21 | ToxCast | SIDER | ClinTox | BACE | HIV | MUV | Avg. |
|---|---|---|---|---|---|---|---|---|---|---|---|
| InfoMax | Supervised | GraphSAGE | 69.12 | 74.17 | 62.65 | 63.22 | 55.43 | 74.70 | 70.44 | 73.60 | 67.92 |
| | | GCN | 68.07 | 74.63 | 59.03 | 63.89 | 55.24 | 63.39 | 76.85 | 71.82 | 66.62 |
| | | GIN | 70.43 | 73.20 | 60.73 | 60.42 | 51.19 | 71.70 | 74.21 | 71.77 | 66.71 |
| | Pre-training+ Fine-tuning | Linear Probing | 66.52 | 78.02 | 66.49 | 65.18 | 73.74 | 84.55 | 77.68 | 80.02 | 74.03 |
| | | Fine Tuning | 69.81 | 78.92 | 66.50 | 66.54 | 71.86 | 82.68 | 76.33 | 81.01 | 74.21 |
| | Prompt Learning | All-In-One | 58.50 | 66.09 | 52.43 | 46.09 | 58.98 | 69.69 | 48.08 | - | 57.12 |
| | | GPF | 67.33 | 77.53 | 65.91 | 65.46 | 73.59 | 83.27 | 74.89 | 79.96 | 73.49 |
| | | GPF-Plus | 67.61 | **79.67** | 65.78 | 64.96 | 72.17 | 81.41 | 71.68 | 78.61 | 72.74 |
| | | IA-GPL | **68.86** | 78.95 | **66.58** | **66.16** | **78.90** | 85.08 | 75.90 | 82.25 | **75.34** |
| AttrMasking | Supervised | GraphSAGE | 71.68 | 73.94 | 61.96 | 62.16 | 61.01 | 63.86 | 73.90 | 76.27 | 68.10 |
| | | GCN | 67.02 | 74.47 | 60.83 | 61.88 | 56.21 | 70.82 | 75.55 | 73.20 | 67.50 |
| | | GIN | 66.43 | 73.69 | 60.98 | 60.29 | 56.65 | 79.63 | 73.48 | 72.75 | 67.99 |
| | Pre-training+ Fine-tuning | Linear Probing | 66.56 | 79.37 | 66.15 | 67.65 | 74.52 | 86.61 | 78.55 | 81.34 | 75.09 |
| | | Fine Tuning | 67.51 | 78.66 | 67.33 | 65.16 | 74.68 | 80.73 | 78.31 | 77.22 | 73.70 |
| | Prompt Learning | All-In-One | 49.79 | 52.78 | 68.26 | 49.57 | 41.69 | 53.46 | 34.97 | - | 50.07 |
| | | GPF | 67.70 | 79.16 | 66.75 | 66.39 | 72.24 | 85.82 | 77.51 | 79.08 | 74.33 |
| | | GPF-Plus | 67.73 | 78.42 | 67.95 | 68.13 | 73.02 | 84.08 | 78.08 | 84.11 | 75.19 |
| | | IA-GPL | **69.35** | **79.30** | **68.52** | **69.66** | **80.15** | **86.78** | **78.90** | **84.70** | **77.17** |
| ContextPred | Supervised | GraphSAGE | 64.12 | 72.05 | 60.20 | 61.99 | 72.94 | 77.77 | 74.18 | 75.85 | 69.89 |
| | | GCN | 63.58 | 71.40 | 62.98 | 57.65 | 70.60 | 79.84 | 78.09 | 75.61 | 70.21 |
| | | GIN | 61.88 | 75.42 | 64.92 | 61.39 | 69.46 | 80.74 | 75.79 | 77.64 | 70.91 |
| | Pre-training+ Fine-tuning | Linear Probing | 65.78 | 80.62 | 59.33 | 65.55 | 70.87 | 78.10 | 76.58 | 83.19 | 72.25 |
| | | Fine Tuning | 67.99 | 78.24 | 63.71 | 63.88 | 73.20 | 81.90 | 79.71 | 81.41 | 73.75 |
| | Prompt Learning | All-In-One | 55.93 | 62.18 | 61.62 | 45.91 | 59.19 | 48.01 | 39.10 | - | 53.13 |
| | | GPF | 67.35 | 78.24 | **68.98** | 63.25 | 70.78 | **83.32** | 78.60 | 82.60 | 74.14 |
| | | GPF-Plus | 68.05 | 77.17 | 68.57 | 64.95 | 75.83 | 81.06 | 76.34 | 85.12 | 74.63 |
| | | IA-GPL | **69.92** | **80.49** | 68.18 | **66.07** | **77.30** | 82.62 | **79.90** | **85.53** | **76.07** |
| GCL | Supervised | GraphSAGE | 67.88 | 68.79 | 63.79 | 51.08 | 72.47 | 68.41 | 71.10 | 68.50 | 66.50 |
| | | GCN | 65.20 | 66.88 | 61.50 | 55.54 | 74.72 | 65.86 | 74.90 | 73.09 | 67.21 |
| | | GIN | 67.56 | 68.65 | 67.14 | 51.17 | 75.79 | 71.06 | 69.07 | 70.69 | 67.73 |
| | Pre-training+ Fine-tuning | Linear Probing | 71.82 | 74.81 | 60.89 | 58.75 | 76.92 | 65.49 | 74.02 | 73.91 | 69.83 |
| | | Fine Tuning | 69.90 | 72.56 | 63.17 | 56.26 | 74.64 | 68.20 | 73.89 | 75.73 | 69.41 |
| | Prompt Learning | All-In-One | 61.07 | 47.33 | 49.54 | 41.07 | 57.70 | 54.24 | 46.17 | - | 50.87 |
| | | GPF | 70.54 | 73.19 | **61.08** | 61.77 | 72.10 | 67.53 | 73.61 | 74.92 | 69.34 |
| | | GPF-Plus | 70.94 | 73.70 | 60.90 | 62.48 | 71.54 | 70.62 | **76.84** | 77.07 | 70.51 |
| | | IA-GPL | **72.58** | **76.08** | 60.86 | **64.22** | **75.07** | **71.24** | 75.59 | **77.33** | **71.62** |

Table 9: The hyperparameter settings for 50-shot learning.

| Dataset | split | Learning rate | #Codebook vectors | #Samples | #MLP layers (Proj. head) |
|---|---|---|---|---|---|
| BBBP | Scaffold | 0.005 | 20 | 10 | 3 |
| Tox21 | Scaffold | 0.0005 | 50 | 10 | 3 |
| ToxCast | Scaffold | 0.0001 | 50 | 10 | 4 |
| SIDER | Scaffold | 0.005 | 10 | 5 | 2 |
| ClinTox | Scaffold | 0.0001 | 50 | 10 | 4 |
| BACE | Scaffold | 0.0001 | 20 | 10 | 2 |
| HIV | Scaffold | 0.005 | 20 | 10 | 4 |
| MUV | Scaffold | 0.0005 | 20 | 10 | 2 |
| BBBP | Random | 0.001 | 20 | 50 | 4 |
| Tox21 | Random | 0.001 | 20 | 50 | 2 |
| ToxCast | Random | 0.005 | 50 | 5 | 2 |
| SIDER | Random | 0.005 | 50 | 5 | 4 |
| ClinTox | Random | 0.001 | 20 | 10 | 2 |
| BACE | Random | 0.001 | 50 | 5 | 4 |
| HIV | Random | 0.005 | 50 | 10 | 3 |
| MUV | Random | 0.005 | 20 | 10 | 2 |

