# OpenReview forum: "Instance-Aware Graph Prompt Learning"
_ICLR.cc/2025/Conference — ICLR 2025 Conference Withdrawn Submission_

### Official Review · Reviewer_XvDk · 2024-11-01

**Soundness:** 3
**Presentation:** 3
**Contribution:** 2
**Rating:** 6
**Confidence:** 4

**Summary:**

Prompt learning has been a hot topic in recent years. This paper explored instance-aware prompt learning for the graph domain. The basic idea is to use fewer parameters while producing more diversity for instance (node) representation. To this end, the paper tried to feed the representations of the input instance into parameterized hypercomplex multiplication (PHM) layers, which transform the feature space into the prompt space with minimal parameters. The vector quantization is further applied to improve prompt diversity.
The experiments on molecular datasets and citation datasets demonstrate its effectiveness.

**Strengths:**

1. The instance-level prompt in the graph domain is interesting and has practical applications like molecules graph modeling.   The proposed prompt method gets competitive performance on molecular prediction tasks, demonstrating its potential in the biomedical field.
2. The design of parameterized hypercomplex multiplication for graph prompt learning is novel, significantly reducing the number of learnable parameters.
3. The using of vector quantization makes the node embedding more interpretable.

**Weaknesses:**

The main concerns are:
1. The benefit of the graph-level task is not clear.  In other words, what is the difference between the universal prompt and the instance prompt for graph-level tasks? It is feasible to use an instance prompt for node-level graph tasks, but for graph-level tasks, how to guarantee the advantages of instance graph prompts?
2. The scalability is a little bit weak. Instance prompts need varying model size according to the node number, in my opinion.
3. The Codebook visualization is not very promising, e.g., for the first figure of Figure 4, most vectors have the same clustering, and demonstrate less diversity for node prompts. Could you provide more details for this question, regarding using more than 5 vectors?
4. Lack of discussion of the difference between the graph domain and visual domain for the instance level prompts.
5. Unafir comparisions. The tuning parameters of the proposed model is larger than the baselines like GPF (see table 6) , so it might be unfair for the baselines.

Some presentation issues:
1. line 244. dimention $d^{'}$ -> $d$?
2. I suggest changing the order of Eq (7) and Eq(8).
3. Line 300, what is instance -agonist prompt? The same question arises in Figure 3.
4. Figure 6    - > Table 3

**Questions:**

see weaknesses

---

### Official Review · Reviewer_1awz · 2024-11-02

**Soundness:** 3
**Presentation:** 3
**Contribution:** 2
**Rating:** 5
**Confidence:** 4

**Summary:**

The paper focuses on graph prompt learning by challenging the prevalent assumption that fixed, task-specific prompts suffice across all instances. Particularly, the article introduces an Instance-Aware Graph Prompt Learning (IA-GPL) framework, designed to generate prompts unique to each input instance, enhancing adaptability and generalization. IA-GPL proposes generating instance-specific prompts using a lightweight parameterized architecture that employs vector quantization and exponential moving average for stable training. Many experiments have been provided and results look promising.

**Strengths:**

S1. The paper identifies a critical limitation in existing graph prompt learning methods—the reliance on static, task-specific prompts—and proposes an instance-aware approach.

S2. The Parameterized Hypercomplex Multiplication (PHM) layers in the prompt generator make the model lightweight, addressing the computational concerns commonly associated with instance-specific adjustments in GNNs.

S3. The integration of a codebook with vector quantization not only helps reduce the high variance caused by the PHM layers but also clusters similar hidden prompt representations together to provide the beneficial property of clustering.

**Weaknesses:**

W1. While authors claim that IA-GPL is the first graph prompting method capable of generating instance-specific prompts, this is untrue as GPF-Plus already explores this. Additionally, while the prompt generator is designed for lightweight purpose, total tuning parameters in GPF-Plus (3-12K) is much lower than that in IA-GPL (20K), as shown in Figure 6. Given this,  the performance improvement of IA-GPL over GPF-Plus is minor, particularly for the results reported in Table 1.

W2. While the PHM layers might be more parameter-efficient than FCNs, there lacks a critical experiment to examine whether PHM offer any performance advantage over FCNs.

W3. Some important references in the field are missing, as follows: (1) PRODIGY: Enabling In-context Learning Over Graphs. (2) Virtual Node Tuning for Few-shot Node Classification.

**Questions:**

see weaknesses

---

### Official Review · Reviewer_rLL8 · 2024-11-04

**Soundness:** 3
**Presentation:** 3
**Contribution:** 3
**Rating:** 5
**Confidence:** 4

**Summary:**

The authors present Instance-Aware Graph Prompt Learning (IA-GPL), a graph prompting method that generates specific prompts for each input instance in downstream tasks. Using a parameter-efficient bottleneck architecture, we create intermediate prompts, which are then quantized with trainable codebook vectors and stabilized through an exponential moving average update. Experimental results are used to validate the effectiveness of their method.

**Strengths:**

a. The presentation is good, and it is easy to follow.
b. The idea generating distinct and specific prompts for individual input instances is interesting.

**Weaknesses:**

a. Since the authors aim to generate distinct prompts for each instance, my main concern is whether the proposed method can scale efficiently on large graphs. Additional experiments on larger graph datasets are encouraged.

b. Additionally, based on Table 1, the improvement offered by the proposed method appears limited, which may not sufficiently demonstrate its effectiveness. Could the authors provide some statistical tests?

c. Given that the proposed method generates distinct prompts for each instance, how does its computational complexity compare to that of the baseline methods?  A theoretical analysis of computational complexity is needed to illustrate the efficiency of the proposed method.

**Questions:**

Please address the above concerns.

---

### Official Review · Reviewer_nqMG · 2024-11-08

**Soundness:** 2
**Presentation:** 2
**Contribution:** 2
**Rating:** 5
**Confidence:** 3

**Summary:**

The authors introduce a prompt tuning method to enhance the performance of GNNs by generating instance-specific prompts. Unlike existing techniques that use static prompts, IA-GPL tailors prompts to individual input instances using lightweight architectures and vector quantization. The proposed method aims to bridge the gap between pretext and downstream tasks while maintaining parameter efficiency.

**Strengths:**

- The introduction of instance-aware graph prompts addresses a critical gap in current graph prompt learning approaches by tailoring prompts to individual instances rather than using fixed, task-specific prompts.
Existing methods, such as GPF, typically rely on static prompts that are applied uniformly across all input data within a given task. While this strategy may work in simpler cases, it fails to generalize effectively to diverse, complex instances, especially in scenarios where data structures are highly heterogeneous. The authors' proposed method, IA-GPL, moves beyond this limitation by generating distinct, instance-specific prompts that adapt dynamically to the input data.

- The authors conducted extensive experiments across diverse datasets, including both in-domain and out-of-domain scenarios, with a focus on challenging cases like few-shot learning. The results consistently showed that IA-GPL outperforms state-of-the-art methods in accuracy and efficiency. Its instance-aware approach led to somewhat gains, especially in out-of-distribution tasks, demonstrating robustness and adaptability. The method excelled across various molecular and citation datasets, proving its versatility while efficiently handling complex graph structures.

- The paper includes extensive ablation studies and promises to release code, supporting reproducibility.

**Weaknesses:**

- Despite utilizing a more complex architecture and significantly increasing the number of trainable parameters, IA-GPL does not exhibit a consistently superior performance over GPF-plus. This raises questions regarding whether the added complexity and computational cost translate into a meaningful improvement.  Especially, given the small gap and the high standard deviation, I'm uncertain whether the results are statistically significant.

|  (50-shot)    |  ToxCast      |   SIDER    |  ClinTox     | BACE     |   HIV    |   MUV    | # Tuning parameters|
|-----|    -   |   -   |     -  |    -  |     -  |    -  |   -  |
|  GPF-plus    |  60.85±1.69     |     52.44±0.83 |   73.85±2.15    |   76.02±0.99   |  64.49±1.19     |  59.93±0.83    | 3-12K|
|  IA-GPL   |  61.63±0.40     |     52.85±0.84 |   74.50±0.76    |   76.64±0.83   |   64.60±0.95    |    59.32±1.13  | 20K (167%~667% of GPF-plus)|



- The IA-GPL method bears similarities to the GPF-plus approach, particularly in the way it generates node-level prompts by leveraging a shared basis. However, a side-by-side comparison of these two methods is notably absent in Figure 2, where the inclusion of GPF-plus would have provided a clearer perspective on how IA-GPL differentiates itself. Also, what are the potential theoretical advantages of IA-GPL compared to GPF-plus? The authors argue that while GPF-plus uses node-specific prompts, IA-GPL is superior; however, the experimental results do not strongly support this claimed superiority.



- Additionally, while IA-GPL incorporates more sophisticated mechanisms and employs additional training parameters, it also incurs greater computational overhead compared to the simpler aggregation technique used in GPF-plus.

**Questions:**

-

---

### Note · Authors · 2024-11-21

I have read and agree with the venue's withdrawal policy on behalf of myself and my co-authors.